# Correlation-Driven Multi-Modality Graph Decomposition for Cross-Subject Emotion Recognition

### Wuliang Huang
Beijing Key Laboratory of Mobile Computing and Pervasive Device, Institute of Computing Technology, Chinese Academy of Sciences, University of Chinese Academy of Sciences
Beijing, China
huangwuliang19b@ict.ac.cn

### Yiqiang Chen*
Beijing Key Laboratory of Mobile Computing and Pervasive Device, Institute of Computing Technology, Chinese Academy of Sciences, Peng Cheng Laboratory, University of Chinese Academy of Sciences
Beijing, China
yqchen@ict.ac.cn

### Xinlong Jiang
Beijing Key Laboratory of Mobile Computing and Pervasive Device, Institute of Computing Technology, Chinese Academy of Sciences, University of Chinese Academy of Sciences
Beijing, China
jiangxinlong@ict.ac.cn

### Chenlong Gao
Beijing Key Laboratory of Mobile Computing and Pervasive Device, Institute of Computing Technology, Chinese Academy of Sciences, University of Chinese Academy of Sciences
Beijing, China
gaochenlong@ict.ac.cn

### Qian Chen
Beijing Key Laboratory of Mobile Computing and Pervasive Device, Institute of Computing Technology, Chinese Academy of Sciences, University of Chinese Academy of Sciences
Beijing, China
chenqian20b@ict.ac.cn

### Teng Zhang
Beijing Key Laboratory of Mobile Computing and Pervasive Device, Institute of Computing Technology, Chinese Academy of Sciences, University of Chinese Academy of Sciences
Beijing, China
zhangteng19s@ict.ac.cn

### Bingjie Yan
Beijing Key Laboratory of Mobile Computing and Pervasive Device, Institute of Computing Technology, Chinese Academy of Sciences, University of Chinese Academy of Sciences
Beijing, China
yanbingjie22s@ict.ac.cn

### Yifan Wang
Tsinghua Shenzhen International Graduate School, Tsinghua University
Shenzhen, China
yifan-wa22@mails.tsinghua.edu.cn

### Jianrong Yang
Institute of Health Management, Guangxi Academy of Medical Sciences, the People's Hospital of Guangxi Zhuang Autonomous Region
Guangxi, China
gandansurgery2014@163.com

## Abstract

Multi-modality physiological signal-based emotion recognition has attracted increasing attention as its capacity to capture human affective states comprehensively. Due to multi-modality heterogeneity and cross-subject divergence, practical applications struggle with generalizing models across individuals. Effectively addressing both issues requires mitigating the gap between multimodal signals while acquiring generalizable representations across subjects. However, existing approaches often handle these dual challenges separately, resulting in suboptimal generalization. This study introduces a novel framework, termed *Correlation-Driven Multi-Modality Graph Decomposition (CMMGD)*. The proposed CMMGD initially captures adaptive cross-modal correlations. It connects each unimodal graph to a multimodal mixed graph. To simultaneously address the dual challenges, it incorporates a correlation-driven graph decomposition module that decomposes the mixed graph into concordant and discrepant subgraphs based on the correlations. The decomposed concordant subgraph encompasses consistently activated features across modalities and subjects during emotion elicitation, unveiling a generalizable subspace. Additionally, we design a Multi-Modality Graph Regularized Transformer (MGRT) backbone specifically tailored for multimodal physiological signals. The MGRT can alleviate the over-smoothing issue and mitigate over-reliance on any single modality. Extensive experiments demonstrate that CMMGD outperforms the state-of-the-art methods by 1.79% and 2.65% on DEAP and MAHNOB–HCI datasets, respectively, under the leave-one-subject-out cross-validation strategy.

*Corresponding author.

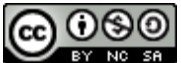

*MM '24, October 28-November 1, 2024, Melbourne, VIC, Australia*
© 2024 Copyright held by the owner/author(s).
ACM ISBN 979-8-4007-0686-8/24/10
https://doi.org/10.1145/3664647.3681579

## CCS Concepts

• **Information systems → Multimedia information systems**;
• **Applied computing → Health informatics**.

## Keywords

Multi-modality, Physiological signal, Graph decomposition, Graph transformers, Emotional state recognition

**ACM Reference Format:**
Wuliang Huang, Yiqiang Chen, Xinlong Jiang, Chenlong Gao, Qian Chen, Teng Zhang, Bingjie Yan, Yifan Wang, and Jianrong Yang. 2024. Correlation-Driven Multi-Modality Graph Decomposition for Cross-Subject Emotion Recognition. In *Proceedings of the 32nd ACM International Conference on Multimedia (MM '24), October 28-November 1, 2024, Melbourne, VIC, Australia.* ACM, New York, NY, USA, 10 pages. https://doi.org/10.1145/3664647.3681579

## 1 Introduction

Multimodal physiological signals, such as electroencephalography (EEG) and peripheral physiological signals (PPS), reflect the cognitive processes of the human [11, 62]. Recognizing emotions from these modalities has attracted increasing attention for various scenarios [15, 67] since humans find it hard to conceal genuine emotions reflected by these signals [61]. Although these signals have been widely used in emotion recognition, capturing generalizable multimodal patterns across diverse subjects remains a grand challenge which hinders their practical applications in real life [29, 48]. **Principal Challenges**. As depicted in Figure 1(a), the distributions of different modalities within the same individual are inconsistent, highlighting the primary aspect of intrinsic *multi-modality heterogeneity*. Furthermore, Figure 1(b) demonstrates that the distributions of multi-modality signals are inconsistencies across individuals, defining the secondary aspect of *cross-subject divergence*.

To effectively tackle these challenges, bridging the gap between multi-modality signals and establishing a generalizable representation across individuals is crucial. Unfortunately, the coupled nature of these dual challenges exacerbates the complexity of devising isolated solutions. As a result, singular approaches aimed at addressing either the heterogeneity or the divergence fail in the context of cross-subject emotion recognition utilizing multimodal physiological signals, leading to suboptimal performance.

Previous studies have primarily relied upon shared representational spaces to obtain subject-independent features for cross-subject scenarios, including robust feature decomposition [10, 25, 50], and selection [29, 66, 76]. However, these approaches necessitate specialized expertise and may not be optimal for diverse tasks. Prior or late fusion of multimodal signals through deep networks have shown progress [2, 5, 20], but cannot wholly resolve issues of multi-modality heterogeneity and cross-subject divergence, limiting their generalizability. Moreover, they also fail to fully leverage the inherent structural information within physiological signals. Recently, transfer learning-based approaches concentrate on generalizability [21–24]. Nevertheless, most methods necessitate calibration data from the target subject, often unavailable in real-world scenarios. Overall, simultaneously and effectively addressing multi-modality heterogeneity and cross-subject divergence remains an open challenge that this work aims to tackle.

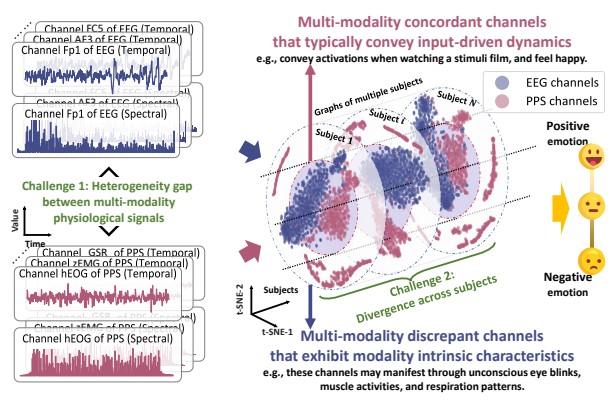

(a) Multimodal signals.  (b) The proposed multimodal graph decomposition.

**Figure 1: An illustration of the dual challenges: multi-modality heterogeneity and cross-subject divergence, and the proposed multi-modality graph decomposition method.**

**The Proposed Solution**. This study introduces a novel unified framework, namely the *Correlation-Driven Multi-Modality Graph Decomposition (CMMGD)*. In detail, the CMMGD initially maps physiological signals within each modality onto graphs. Subsequently, fine-grained adaptive cross-modal correlations between modalities are developed, forming a multi-modality mixed graph.

A pivotal step in this framework involves the decomposition of the mixed graph into concordant and discrepant subgraphs driven by the learned correlations. The concordant subgraph contains channels activated consistently across modalities and subjects during emotion elicitation, thereby delineating a generalizable subspace. Specifically, this subspace is devised to address the primary multimodal heterogeneity while mitigating cross-subject divergence. Additionally, the discrepant subgraph conveys modality-intrinsic activations, such as muscle activity and respiration patterns. Finally, a cross-rebalance fusion mechanism is devised to fuse features from the concordant and discrepant subgraphs in a balanced manner, realizing precise emotional state prediction.

Within the CMMGD framework, we design a novel backbone specifically for multimodal physiological signals, termed the Multi-Modality Graph Regularized Transformer (MGRT). More precisely, the MGRT incorporates a strategy of localized graph regularization, which is applied in parallel with global multi-modality attention. Such a concurrent methodology effectively addresses the problem of over-smoothing [26] —a known challenge arising from the sequential application of graph convolution and attention layers [4, 14, 26], as well as issues related to small graph scales in physiological signals [18, 49]. Furthermore, integrating local and global features minimizes the risk of excessive reliance on any modality and in turn, significantly augments the generalizability.

**Contributions**. The principal contributions are detailed as follows:

(1) We propose a novel multi-modality correlation-driven graph decomposition module to learn a generalizable space that simultaneously addresses the dual challenges of multi-modality heterogeneity and cross-subject divergence.

(2) We develop a novel MGRT backbone specifically for multimodal physiological signals, mitigating the over-smoothing issue and avoiding over-reliance on any single modality, thus further promoting generalizability.

(3) We establish the CMMGD framework to integrate the above innovations. Comprehensive experiments on two benchmark datasets demonstrate the superiority of the CMMGD framework over the state-of-the-art methods.

## 2 Related Work

Physiological signal-based emotion recognition has a longstanding history [33, 41, 42], aiming to identifying human affective states using the dimensional model [46], which conceptualizes emotions along the dimensions of arousal and valence. The valence dimension describes whether an emotion is positive or negative, whereas arousal refers to its intensity.

**Multi-modality Emotion Recognition**. Previous studies have synergistically integrated multi-modal data like EEG and PPS to enhance emotion recognition performance [39, 68]. Compared to unimodal approaches, these multimodal methods have shown superior performance [32, 37, 53, 74]. Existing fusion methods adopt the concatenation or attention mechanisms to combine features from different modalities [8, 55, 64]. However, these methods do not explicitly address the correlations among modalities and the variations within each modality, potentially leading to suboptimal performance [12]. To learn inter-modality relationships, correlation-based fusion [70, 71], canonical correlation analysis [73] and graph-based models [13] have been proposed. However, their characterizations of cross-modal correlations remain coarse. The proposed CMMGD represents each modality as an unimodal graph, capturing adaptive and fine-grained cross-modal correlation to build a mixed graph. It further decomposes this graph into concordant and discrepant subgraphs, providing a more precise representation.

**Cross-Subject Emotion Recognition**. There are two basic validation strategies: subject-dependent and subject-independent. The former is more common and has shown better performance [8, 55, 64]. The latter requires generalization across subjects, remaining a challenge [2, 5, 20]. Robust feature selection methods, including feature decomposition [10, 25, 50] and channel selection [29, 66, 76] partially mitigate cross-subject divergence but require domain expertise. Recently, transfer learning improves generalizability [21–24]. However, most of them operate within domain adaptation [19], which necessitates calibration data from the target subject. Moreover, the above methods fail to address multi-modality heterogeneity explicitly. In comparison, the proposed CMMGD framework reveals a generalizable subspace across modalities and subjects without additional target data, providing a unified solution to multimodal heterogeneity and cross-subject divergence.

**Graph Transformer Architecture**. Representing physiological signals as graphs have gained popularity since graph structures can preserve natural spatial and functional connectivity among electrodes [13, 43, 65]. The graph-based methods serve as the foundation [51]. Among these, graph transformers are recent advancements that have shown promise in capturing dependencies within and across modalities [4, 47, 63]. Rampasek et al. [44] introduce a robust and versatile graph transformer with linear complexity.

Jiang et al. [14] propose a graph transformer specifically for emotion recognition. Nevertheless, the utilization of small-scale graphs in emotion recognition [18, 49] exacerbates the challenge of over-smoothing [26], a concern that most existing graph transformer methodologies do not explicitly address. We introduce a Multi-Modality Graph Regularized Transformer (MGRT) backbone designed to mitigate over-smoothing while enhancing generalizability.

## 3 Preliminaries

**Problem Formulation**. The CMMGD model $\mathcal{M}_{\text{CMMGD}}$ aims to predict emotional states leveraging multimodal physiological signals, specifically, EEG and PPS: $\widehat{\mathcal{Y}} = \mathcal{M}_{\text{CMMGD}}(X_e, X_p)$. The notation $\widehat{\mathcal{Y}}$ signifies the emotional states on the valence or arousal dimension. The pair $(X_e, X_p)$ alludes to a multimodal sample, where $X_e \in \mathbb{R}^{c_e \times T}$ and $X_p \in \mathbb{R}^{c_p \times T}$ denote the EEG and PPS data. Here, $c_e$ and $c_p$ represent the number of EEG and PPS channels, while $T$ signifies the temporal duration. To enhance clarity, the subscripts $e$ and $p$ in the following paragraphs specifically pertain to EEG and PPS, respectively.

**Theoretical Insights**. In the domain of cross-subject emotion recognition, involving $K + 1$ subjects, it requires to adopt the leave-one-subject-out (LOSO) cross-validation strategy [16], which trains the model on $K$ visible subjects with set of distributions $\{\mathbb{P}_i | i = 1, \ldots, K\}$, and validates it on the left-out $\mathbb{Q}$. We aim at minimizing the error on the left-out subject $\varepsilon_{\mathbb{Q}}(h)$, by leveraging the distributions of visible subjects. To this end, we provide theoretical insights into the generalization error of cross-subject emotion recognition:

**THEOREM 3.1.** *Let $\mathbb{Q}$ and $\{\mathbb{P}_i | i = 1, \ldots, K\}$ be distributions over space $X$, $\mathcal{H}$ be a class of hypotheses corresponding to this space, and $\{\varphi_i\}_{i=1}^{K}$ be a collection of non-negative coefficients with $\sum_i \varphi_i = 1$. Let $O$ be a set of distributions, such that for every $\mathbb{S} \in O$, we have:*

$$\sum_i \varphi_i d_{\mathcal{H} \Delta \mathcal{H}}(\mathbb{P}_i, \mathbb{S}) \leq \max_{i,j} d_{\mathcal{H} \Delta \mathcal{H}}(\mathbb{P}_i, \mathbb{P}_j). \tag{1}$$

*Then, for any $h \in \mathcal{H}$:*

$$
\begin{aligned}
\varepsilon_{\mathbb{Q}}(h) \leq \lambda_\varphi + \sum_i \varphi_i \varepsilon_{\mathbb{P}_i}(h) \\
+ \frac{1}{2} \min_{\mathbb{S} \in O} d_{\mathcal{H} \Delta \mathcal{H}}(\mathbb{S}, \mathbb{Q}) + \frac{1}{2} \max_{i,j} d_{\mathcal{H} \Delta \mathcal{H}}(\mathbb{P}_i, \mathbb{P}_j),
\end{aligned}
\tag{2}
$$

*where the $\varepsilon_{\mathbb{Q}}(h)$ is the error for a hypothesis $h$ on the left-out subject, $\lambda_\varphi$ is the error of an ideal joint hypothesis which could be neglected. $d_{\mathcal{H} \Delta \mathcal{H}}(\mathbb{P}, \mathbb{Q})$ is $\mathcal{H}$-divergence which measures the difference between two distributions. The proof of this theorem has been given in [1, 36].*

The above Theorem 3.1 explores the generalization error $\varepsilon_{\mathbb{Q}}(h)$ of the cross-subject emotion recognition. The reduction of the second term $\sum_i \varphi_i \varepsilon_{\mathbb{P}_i}(h)$ can be accomplished through supervised emotional state recognition loss $\mathcal{L}_{\text{emo}}$ introduced in Section 4.3. The last term $\frac{1}{2} \max_{i,j} d_{\mathcal{H} \Delta \mathcal{H}}(\mathbb{P}_i, \mathbb{P}_j)$ entails aligning the distributions from visible subjects. Note that, we focus on multimodal signals where $\mathbb{P}$ is the distribution of multimodal samples. We leverage an alignment loss $\mathcal{L}_{\text{aln}}$ introduced in Section 4.2 to jointly align multimodal features, thus minimizing the last term in (2).

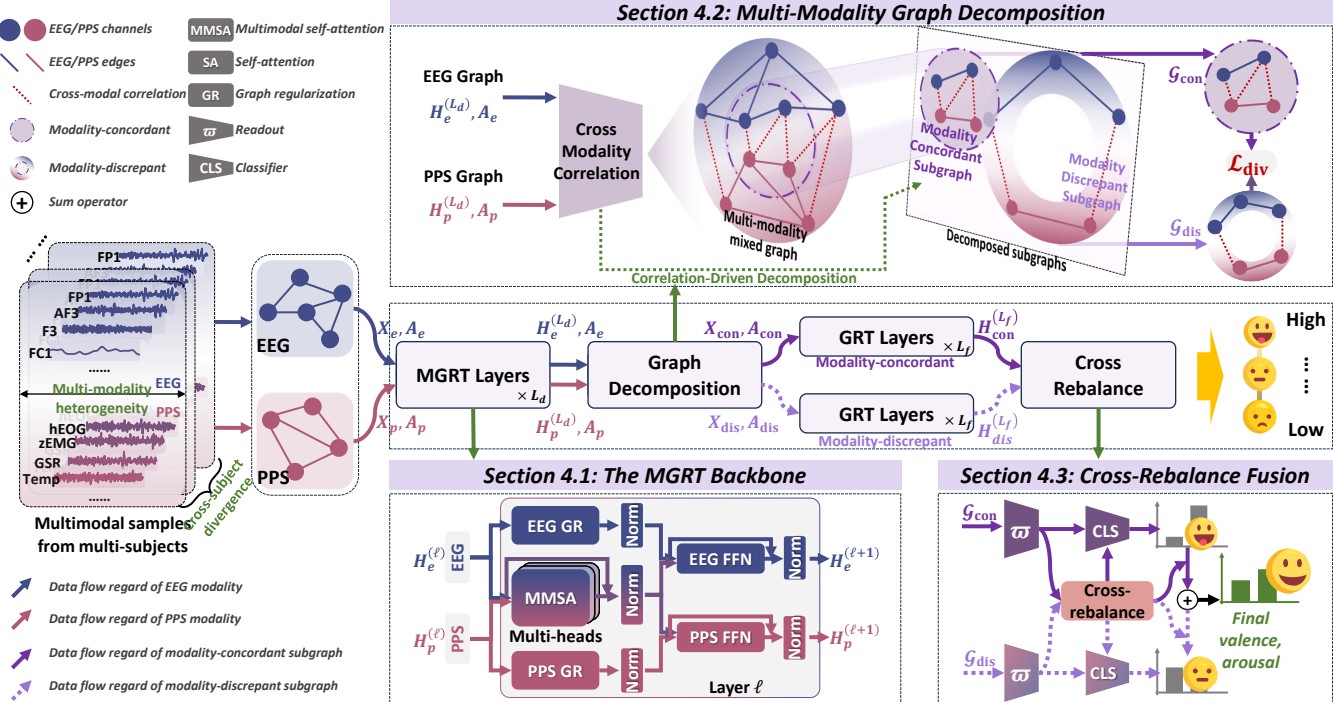

**Figure 2: The architecture of the CMMGD framework for cross-subject multimodal emotion recognition comprises three main components: (1) the MGRT backbone, (2) the graph decomposition module, and (3) the cross-rebalance fusion module.**

In the subsequent phase, taking into account prior works [1, 7], and (1), it is essential to acquire a diverse distribution of visible subjects to minimize the third term $\frac{1}{2} \min_{\mathbb{S} \in \mathcal{O}} d_{\mathcal{H}\Delta\mathcal{H}}(\mathbb{S}, \mathbb{Q})$. We propose achieving this by learning diverse concordant and discrepant representations of multimodal physiological signals using an auxiliary diverse loss $\mathcal{L}_{\text{div}}$ detailed in Section 4.2.

## 4 The proposed CMMGD Framework

The proposed *Correlation-Driven Multi-Modality Graph Decomposition (CMMGD)* framework is designed to jointly address the dual challenges of multi-modality heterogeneity and cross-subject divergence. We subsequently detail each component in conjunction with the overall architecture in Figure 2.

### 4.1 The MGRT Backbone

**Encoding Spatial and Functional Connectivity**. The MGRT initially represents each modality as graphs [13, 64], and further captures both intra- and inter-modality dependencies. Two types of connection encoding are considered to maintain prior EEG and PPS structures: (1) **Spatial encoding**: It is established based on the physical proximity of the electrodes within the standard 10-20 system of EEG [18]. The linear distances $\Omega \in \mathbb{R}^{c \times c}$ between $c$ electrodes can be computed as $\Omega_{i,j} = \|\omega_i - \omega_j\|_2$, where $\omega \in \mathbb{R}^{c \times 3}$ denotes the 3D coordinates of the electrodes. (2) **FC encoding**: The functional connection is constructed using the mutual information [12, 13], which captures both linear and non-linear relationships. The functional connection matrix $\Phi \in \mathbb{R}^{c \times c}$ is determined

by $\Phi_{i,j} = \sum_{m \in X_i} \sum_{n \in X_j} \log \frac{p(m,n)}{p(m)p(n)}$. We form the EEG adjacency matrix $A_e = \lambda_\Omega \Omega + (1 - \lambda_\Omega)\Phi$, where $\lambda_\Omega$ is a hyperparameter. For PPS, only the functional connection is utilized $A_p = \Phi$, since the spatial encoding is not applicable. To ensure sparsity and robustness, we further reserve merely the top 30% strong connections.

**Embedding Temporal Dynamics**. Deriving from previous study [28], one-dimensional convolutional network $f^t$ is incorporated to capture the temporal patterns. The temporal features are $H_e^{(0)} \in \mathbb{R}^{c_e \times d}$ and $H_p^{(0)} \in \mathbb{R}^{c_p \times d}$, where $d$ is the hidden size.

**Multi-Modality Self-Attention (MMSA)**. The self-attention mechanism is employed to capture the inter-modality global dependencies between EEG and PPS. Let $H_{ep}^{(\ell-1)} = \left[H_e^{(\ell-1)} \| H_p^{(\ell-1)}\right] \in \mathbb{R}^{(c_e+c_p) \times d}$ represent the stacked hidden features at the $(\ell-1)$-th layer. $[\cdot \| \cdot]$ signifies concatenation. The $\ell$-th multi-heads MMSA is computed following Vaswani et al. [59] with output $H_{ep}^{(\ell)}$.

**Intra-Modality Graph Regularization (GR)**. Incorporating recent advancements in graph transformers facilitates the comprehensive perception of both localized and global features within and across modalities. The graph transformer integrates graph convolution and self-attention layers sequentially [14, 56]. However, this sequential application is not optimal for small-scale graphs derived from physiological signals. To explain, We revisit the structure of the typical graph convolution layer [17]. A typical GCN layer is $H_{\text{gcn}} = \sigma\left(\tilde{D}^{-\frac{1}{2}} \tilde{A} \tilde{D}^{-\frac{1}{2}} H W_{\text{gcn}}\right)$, where the normalized adjacent matrix $\tilde{A} = A + I$, the degree matrix $\tilde{D}_{ii} = \sum_j \tilde{A}_{ij}$, and

$\sigma$ denotes the activation function. The matrix $W_{\text{gcn}} \in \mathbb{R}^{d \times d}$ signifies the linear transformation. For small-scale graphs, the GCN layer might oversmooth features [26] or excessively suppress local features, creating a bottleneck. The sequential application of GCN and self-attention layers further exacerbates these issues as the self-attention layer partially smooths the features.

To address these issues, we devise a novel GR operation, where the $\ell$-th GR operation for EEG is defined as:

$$H_{e,\text{gr}}^{(\ell)} = \text{GR}\left(H_e^{(\ell-1)}, A_e\right) = \sigma\left(\tilde{D}_e^{-\frac{1}{2}} \tilde{A}_e \tilde{D}_e^{-\frac{1}{2}} H_e^{(\ell-1)}\right). \quad (3)$$

The PPS graph regularization operation is defined analogously, yielding output $H_{p,\text{gr}}^{(\ell)}$. Subsequently, GR and MMSA are simultaneously employed to derive the final output of the $\ell$-th MGRT layer as:

$$H_e^{(\ell)} = \sigma\left(f_e^{\Theta_{\text{ffn}}}\left(H_{e,\text{gr}}^{(\ell)} + H_{e,\text{att}}^{(\ell)}\right)\right), \; H_p^{(\ell)} = \sigma\left(f_p^{\Theta_{\text{ffn}}}\left(H_{p,\text{gr}}^{(\ell)} + H_{p,\text{att}}^{(\ell)}\right)\right), \quad (4)$$

where $f_e^{\Theta_{\text{ffn}}}$ and $f_p^{\Theta_{\text{ffn}}}$ are feed-forward layers for EEG and PPS. Different GRs are applied for EEG and PPS to prevent overreliance on any single modality, thereby enhancing model generalizability. The parallel operation of GR and MMSA helps alleviate the bottleneck issue, with the GR operation serving as a regularizer for global MMSA, ensuring a balanced fusion of local and global features. With a total of $L_d$ layers, the final vertex features of the MGRT backbones are $H_e^{(L_d)}$, and $H_p^{(L_d)}$.

## 4.2 Multi-Modality Graph Decomposition

To further reveal a generalizable space that can address the multi-modality heterogeneity and cross-subject divergence, this section elucidates the decomposition of multimodal physiological signals into concordant and discrepant subgraphs driven by correlations.

**Learning Multi-modality Mixed Graph**. We first construct a mixed graph to connect EEG and PPS modalities, aiming at capturing intricate adaptive cross-modal relationships. $\Gamma \in \mathbb{R}^{c_e \times c_p \times 2}$ denotes cross-modal correlation between the $i$-th EEG and the $j$-th PPG channel, where each $\Gamma_{ij} \sim p\left(\Gamma_{ij}\big|H_{e,i}^{(L_d)}, H_{p,j}^{(L_d)}\right)$ is sampled from their vertex features:

$$\Gamma_{ij} = \text{Softmax}\left(\frac{p_{ij} + g}{\tau}\right), \text{ where } p_{ij} = \left[H_{e,i}^{(L_d)}\big\|H_{p,j}^{(L_d)}\right] W_c + b_c. \quad (5)$$

The $W_c \in \mathbb{R}^{2d \times 2}$, $b_c \in \mathbb{R}^2$ are learnable parameters. $g \in \mathbb{R}^2$ is a vector of independent and identically distributed (i.i.d.) samples drawn from a standard Gumbel distribution, and $\tau$ is the temperature parameter that governs the smoothness of $\Gamma$. The entire correlation process is differentiable [12, 38]. We adopt a curriculum learning approach by gradually annealing $\tau$ after each epoch to facilitate the convergence [69].

We assign the first dimension of $\Gamma_{ij}$ to signify the presence of correlation between the $i$-th EEG channel and the $j$-th PPS channel, denoted as $Z_{ij}$, and the cross-modal correlation matrix $Z \in \mathbb{R}^{n_e \times n_c}$. Larger values indicate stronger cross-modal correlation.

The multi-modality mixed graph $\mathcal{G}_{\text{mixed}}$ amalgamates the EEG and PPS graphs based on their correlation matrix $Z$. $\mathcal{G}_{\text{mixed}}$ consists of $c_e + c_p$ vertices with features $X_{\text{mixed}} = \left[H_e^{(L_d)}\big\|H_p^{(L_d)}\right] \in \mathbb{R}^{(c_e + c_p) \times d}$. The adjacency matrix $A_{\text{mixed}} \in \mathbb{R}^{(c_e + c_p) \times (c_e + c_p)}$ is constructed by concatenating $A_e$, $A_p$, and $Z$ as described above.

**Correlation-Driven Channel Ranking**. As elucidated in Section 1, the primary challenges in cross-subject multimodal emotion recognition arise from the multi-modality heterogeneity and cross-subject divergence. We aim to address these challenges concurrently by decomposing the mixed graph into concordant and discrepant subgraphs driven by the cross-modal correlations.

The decomposition commences by ranking the channels. We assess the overall significance of each channel with all other channels in the opposing modality, represented as $\xi_{e,i} = \sum_j Z_{ij}$ and $\xi_{p,j} = \sum_i Z_{ij}^{\text{T}}$, where $\xi_{e,i}$ and $\xi_{p,j}$ denote the score of the $i$-th EEG and the $j$-th PPS channel, respectively, with $\xi_e \in \mathbb{R}^{c_e}$ and $\xi_p \in \mathbb{R}^{c_p}$. The top-$\rho$ channels are selected for the concordant subgraph, while the remaining channels are categorized as discrepant channels:

$$\text{idx}_{\text{con}} = \left[\text{argsort}(\xi_e)_{1:\rho c_e}\big\|\text{argsort}(\xi_p)_{1:\rho c_p}\right],$$
$$\text{idx}_{\text{dis}} = \left[\text{argsort}(\xi_e)_{\rho c_e+1:c_e}\big\|\text{argsort}(\xi_p)_{\rho c_p+1:c_p}\right], \quad (6)$$

where $\text{idx}_{\text{con}}$ denotes the index of the concordant channels, while $\text{idx}_{\text{dis}}$ pertains to the discrepant channels. Here, $\rho$ functions as a hyperparameter.

**Concordant and Discrepant Subgraph Decomposition**. The concordant subgraph $\mathcal{G}_{\text{con}}$ and the discrepant subgraph $\mathcal{G}_{\text{dis}}$ are subsequently derived from $\mathcal{G}_{\text{mixed}}$. The adjacency matrices $A_{\text{con}}$ and $A_{\text{dis}}$ can be obtained from $A_u$ by extracting the rows and columns corresponding to the concordant and discrepant subgraphs. Similarly, the features $X_{\text{con}}$ and $X_{\text{dis}}$ can be derived from $X_u$.

$$X_{\text{con}} = X_u\left[\text{idx}_{\text{con}}, :\right], \quad A_{\text{con}} = A_u\left[\text{idx}_{\text{con}}, \text{idx}_{\text{con}}\right],$$
$$X_{\text{dis}} = X_u\left[\text{idx}_{\text{dis}}, :\right], \quad A_{\text{dis}} = A_u\left[\text{idx}_{\text{dis}}, \text{idx}_{\text{dis}}\right], \quad (7)$$

where $[\cdot, \cdot]$ denotes the row and column selection by indices. The whole decomposition process is described with pictures in Figure 2, where the outside circle represents the discrepant subgraph, while the circular area inside represents the concordant subgraph.

**The Divergence Loss**. A divergence loss is introduced to promote the acquisition of diverse representations, as elaborated in Section 3 The distance correlation (dCor) [52, 75] is utilized since it can quantify the dependencies between the concordant and discrepant subgraphs with no assumption of linearity or normality. We randomly sample $n_s$ vertices from the concordant and discrepant subgraphs, $n_s \leq \min\left(c_e, c_p\right)$. The features of the sampled vertices are denoted as $S_{\text{con}}, S_{\text{dis}} \in \mathbb{R}^{n_s \times d}$ respectively. The divergence loss $\mathcal{L}_{\text{div}}$ is:

$$\mathcal{L}_{\text{div}} = \frac{\mathcal{V}_{n_s}^2\left(S_{\text{con}}, S_{\text{dis}}\right)}{\sqrt{\mathcal{V}_{n_s}^2\left(S_{\text{con}}, S_{\text{con}}\right) \mathcal{V}_{n_s}^2\left(S_{\text{dis}}, S_{\text{dis}}\right) + \epsilon}}. \quad (8)$$

$\mathcal{V}_{n_s}^2\left(S_{\text{con}}, S_{\text{dis}}\right)$ is the empirical distance covariance. $\mathcal{V}_{n_s}^2\left(S_{\text{con}}, S_{\text{con}}\right)$ and $\mathcal{V}_{n_s}^2\left(S_{\text{dis}}, S_{\text{dis}}\right)$ are the empirical variances.

**The Alignment Loss**. Furthermore, we follow Section 3 to introduce an auxiliary alignment loss $\mathcal{L}_{\text{aln}}$ aimed at aligning the distributions of the visible subjects. We represent the features of the mixed graph of the $i$-th sample as $X_{\text{mixed}}^{}$. In the model implementation, each batch contains samples from all visible subjects. The alignment loss within a batch $\mathcal{B}$ can be expressed as:

$$\mathcal{L}_{\text{aln}} = \frac{1}{|\mathcal{B}|} \sum_{i=1}^{|\mathcal{B}|} \left\|\text{GMP}\left(X_{\text{mixed}}^{}\right) - \mu\right\|_1, \quad (9)$$

where $\mu = \frac{1}{|\mathcal{B}|} \sum_{j=1}^{|\mathcal{B}|} \text{GMP}\left(X_{\text{mixed}}^{<j>}\right) \in \mathbb{R}^d$. The GMP$(\cdot) \in \mathbb{R}^d$ represents adopting the global mean pooling operation.

## 4.3 Cross-Rebalance Mechanism and Fusion

**Emotional State Prediction**. Naturally, the concordant and discrepant subgraphs are not equally noteworthy since the former provides more generalizable features. A cross-rebalance mechanism is introduced to assess the importance of these two subgraphs precisely, assigning weight factors $\alpha_1$ and $\alpha_2$ to the two subgraphs, respectively. The final prediction of emotional states is given by:

$$R_{\text{con}} = \varpi\left(H_{\text{con}}^{(L_f)}\right), \quad R_{\text{dis}} = \varpi\left(H_{\text{dis}}^{(L_f)}\right). \tag{10}$$

$$\widehat{\mathcal{Y}} = \text{Softmax}\left(\alpha_1 f_{\text{con}}^c\left(\alpha_1 R_{\text{con}}\right) + \alpha_2 f_{\text{dis}}^c\left(\alpha_2 R_{\text{dis}}\right)\right), \tag{11}$$

where $\varpi(\cdot)$ denotes the readout function implemented by the Equilibrium aggregation method [3]. The classifiers $f_{\text{con}}^c$ and $f_{\text{dis}}^c$ consist of two linear layers. The $H_{\text{con}}^{(L_f)}$ and $H_{\text{dis}}^{(L_f)}$ are the final high-level features of the concordant and discrepant subgraphs after $L_f$ feature extractors. We adopt a weak version of MGRT as the feature extractor, where treating the multi-modality channels as a single modality reduces the model complexity, namely the GRT layers.

**Cross-Rebalance Mechanism**. The unresolved issue pertains to the computation of the weight factors $\alpha_1$ and $\alpha_2$ through the cross-rebalance mechanism. We balance theie significance as follows:

$$\psi_1 = \tanh\left(f_{\text{con}}^g\left(R_{\text{con}}\right)\right), \quad \psi_2 = \tanh\left(f_{\text{dis}}^g\left(R_{\text{dis}}\right)\right), \tag{12}$$

where the vector $\psi_1 \in \mathbb{R}^2$ evaluates the importance from the concordant perspective, and $\psi_2 \in \mathbb{R}^2$ does so from the discrepant perspective. $f_{\text{con}}^g$ and $f_{\text{dis}}^g$ are two-layer linear layers. The vector $\alpha = \frac{1}{2}(\psi_1 + \psi_2) \in \mathbb{R}^2$ is the ultimate cross-rebalance weight. The weight factors $\alpha_1$ and $\alpha_2$ are the first and the second dimension of $\alpha$. A cross-rebalance loss $\mathcal{L}_{\text{cross}}$ is introduced to ensure the consistency of the significance assessments between two perspectives:

$$\mathcal{L}_{\text{cross}} = \frac{1}{2}\left(D_{\text{KL}}\left(\psi_1, \psi_2\right) + D_{\text{KL}}\left(\psi_2, \psi_1\right)\right), \tag{13}$$

where $D_{\text{KL}}\left(x, y\right) = \sum_i x_i \log \frac{x_i}{y_i}$ is the Kullback-Leibler divergence.

**Model Training**. The final loss function $\mathcal{L}$ comprises the divergence loss $\mathcal{L}_{\text{div}}$, alignment loss $\mathcal{L}_{\text{aln}}$, cross-rebalance loss $\mathcal{L}_{\text{cross}}$, and supervised emotion recognition loss $\mathcal{L}_{\text{emo}} = -\sum_{i=1}^{|\mathcal{B}|} \mathcal{Y}_i \log \widehat{\mathcal{Y}}_i$, where $\mathcal{Y}$ denotes the ground truth. $\mathcal{L}$ is formulated as:

$$\mathcal{L} = \mathcal{L}_{\text{emo}} + \lambda_{\text{div}}\mathcal{L}_{\text{div}} + \lambda_{\text{aln}}\mathcal{L}_{\text{aln}} + \lambda_{\text{cross}}\mathcal{L}_{\text{cross}}, \tag{14}$$

where $\lambda_{\text{div}}$, $\lambda_{\text{aln}}$, and $\lambda_{\text{cross}}$ denote the hyperparameters.

### Table 1: Dataset Descriptions.

| Dataset | Subject | Modality (channels) | Rate | Total time |
|---|---|---|---|---|
| DEAP | 32 (16 female) | EEG (32), PPS (8) | 128 Hz | 76,800 s |
| MAHNOB–HCI | 27 (16 female) | EEG (32), PPS (6) | 256 Hz | 43,350 s |

## 5 Experimental Evaluation

### 5.1 Datasets and Experimental Setup

Table 1 presents the statistics of the DEAP [18] and MAHNOB–HCI [49] datasets, which are two widely used benchmark multimodal physiological datasets for emotion recognition. We split each trial into 4-second segments with no overlap following [35]. This strategy increases the number of samples and forces the model to learn more robust short-time features. During all the experiments, we leverage the leave-one-subject-out cross-validation.

### 5.2 Comparison Analysis

**Comparison of CMMGD**. Table 2 and Table 3 present the comparison results. From an overarching perspective, the CMMGD framework attains superior performance on both datasets, surpassing the state-of-the-art methods. It excels in nearly all emotion dimensions. These outcomes substantiate the efficacy of the proposed CMMGD framework in handling multi-modality heterogeneity and cross-subject divergence. From the results, methods integrating multimodal signals (Lower part of the Table 2 and 3) lead to superior performance compared to those relying solely on EEG (Middle part of the Table 2 and 3).

The highest overall performance of EEG-only methods is 64.46%, achieved by EEGFuseNet [31]. The multimodal methods demonstrate enhanced performance. Among these multimodal methods, the proposed CMMGD framework showcases the most superior overall performance of 68.59% on DEAP and 66.88% on MAHNOB–HCI, representing improvements of 1.76% and 2.65% over the second-best method, respectively. These enhancements further emphasize the effectiveness of the CMMGD framework.

**Comparison of the MGRT Backbone**. We proceed to evaluate the effectiveness of the proposed MGRT backbone. The comparative results are outlined in Table 4, encompassing two latest graph transformers, namely GraphGPS [44] and EmoGTs [14], as well as two traditional graph-only methods, GCN [17], GraphConv [40], and Transformer [59]. EmoGTs is the latest graph transformer network devised for multimodal emotion recognition.

From Table 4, the graph transformer-based methods outperform graph-only and transformer-only approaches. Among these architectures, our MGRT gains the highest overall performance. This result is due to the parallel design of MMSA and GR, which alleviates the over-smoothing issue and mitigates over-reliance on any single modality, thereby enhancing generalizability.

### 5.3 Ablation Studies

**Ablation Study of Dropping Discrepant Subgraph**. It is acceptable to discard the discrepant subgraph Wang et al. [60] as they convey less consistent information across modalities and subjects. The results of utilizing only the concordant channels are presented in Table 5. The performance decreases when discrepant features are discarded, proving that both the concordant and discrepant features are essential. The concordant features alone are insufficient to capture the full dynamics of emotion.

**Ablation Study of Fusion Strategies**. We introduce a cross-rebalance schema for integrating concordant and discrepant subgraphs. To validate the effectiveness of this fusion approach, we conduct an

**Table 2: Comparison of the proposed CMMGD with the high-level state-of-the-art methods on the DEAP dataset.**

| Method | Publication | Subject Independent | Cross Validation Mode | Arousal | | Valence | | Overall Metrics |
|---|---|---|---|---|---|---|---|---|
| | | | | Accuracy | F1 Score | Accuracy | F1 Score | |
| RBM [48] | ICASSP'17 | | LOTO | 64.6/- | 51.2/- | 60.7/- | 54.1/- | 57.65 |
| LSVM-GSU [57] | TPAMI'18 | | LOTO | 65.9/- | 55.1/- | 65.0/- | 60.9/- | 61.73 |
| MIL [45] | TAFFC'19 | | LOTO | 61.1/- | 54.6/- | 63.6/- | 61.2/- | 60.13 |
| TSception [6] | TAFFC'23 | | LOTO | 63.75/- | 63.35/- | 62.27/- | 65.37/- | 63.64 |
| ACRNN [55] | TAFFC'20 | ✓ | LOSO | 55.00/10.24 | - | 54.84/6.43 | - | 54.92 |
| BiDANN [30] | TAFFC'21 | ✓ | LOSO | 61.04/6.48 | - | 58.70/11.16 | - | 59.87 |
| EEGFuseNet [31] | TNSRE'21 | ✓ | LOSO | 58.55/- | **72.00**/- | 56.44/- | 70.83/- | 64.46 |
| AP-CapsNet [34] | KBS'23 | ✓ | LOSO | 63.51/- | - | 62.71/- | - | 63.11 |
| TMLP+SRDANN [27] | MEASUREMENT'23 | ✓ | LOSO | 57.70/7.23 | - | 61.88/5.55 | - | 59.79 |
| CAFNet [77] | TAFFC'23 | ✓ | LOSO | 62.25/11.47 | 69.28/16.72 | 63.61/9.35 | 61.23/13.87 | 64.09 |
| MMResLSTM [37] | MM'19 | ✓ | LOSO | 63.25/12.38 | 67.32/15.92 | 64.67/10.57 | 68.36/11.50 | 66.15 |
| RDFKM [72] | TCYB'21 | ✓ | LOSO | 63.1/- | 70.1/- | 64.5/- | 69.6/- | 66.83 |
| CSDAMER [9] | BIBM'22 | ✓ | LOSO | 56.85/- | 42.03/- | 62.09/- | 58.00/- | 54.74 |
| EmotionKD [35] | MM'23 | ✓ | LOSO | 62.88/- | 60.23/- | 66.61/- | 66.54/- | 64.07 |
| RHPRNet [54] | INFORM FUSION'24 | ✓ | LOSO | 57.73/3.19 | 59.30/4.64 | 59.42/4.40 | 60.32/4.55 | 59.10 |
| CMMGD (Ours) | | ✓ | LOSO | **64.18**/10.15 | 70.75/16.25 | **66.89**/6.34 | **72.55**/6.81 | **68.59** |

*LOSO means leave-one-subject-out, and LOTO denotes subject-dependent leave-one-trial-out. Values are reported in mean/std format. **Bold** means the best result while underline means the second-best among methods adopting the LOSO cross-validation strategy. The following tables are reported in the same format.

**Table 3: Comparison of the proposed CMMGD with the state-of-the-art methods on the MAHNOB–HCI dataset.**

| Method | Cross Validation Mode | Arousal | | Valence | | Overall Metrics |
|---|---|---|---|---|---|---|
| | | Accuracy | F1 Score | Accuracy | F1 Score | |
| RBM [48] | LOTO | 65.9/- | 65.4/- | 59.1/- | 54.2/- | 61.15 |
| TSception [6] | T-10F | 60.61/14.88 | 33.06/23.35 | 61.27/10.05 | 40.66/16.52 | 48.90 |
| EEGFuseNet [31] | LOSO | 62.06/- | 62.05/- | 60.64/- | 72.18/- | 64.23 |
| CSDAMER [9] | LOSO | 60.47/- | 46.12/- | 62.23/- | 49.64/- | 54.62 |
| EmotionKD [35] | LOSO | 60.66/- | 58.32/- | 64.72/- | 64.27/- | 61.99 |
| CMMGD (Ours) | LOSO | **66.41**/11.00 | **62.61**/23.19 | **65.86**/7.15 | **72.65**/6.77 | **66.88** |

*T-10F means subject-dependent trial-wise ten-fold cross-validation.

ablation study comparing it with three commonly used methods: summation, maximum, and concatenation. The results are presented in Table 6, and the proposed cross-rebalance fusion mechanism exhibits superior performance by adaptive assigning weights to the concordant and discrepant subgraphs.

**Ablation Study of Auxiliary Losses**. We proceed with an ablation study to assess the effectiveness of $\mathcal{L}_{div}$ and $\mathcal{L}_{aln}$, which serve as supplementary losses aimed at enhancing cross-subject generalizability. Table 7 presents the detailed results and the findings demonstrate that omitting either $\mathcal{L}_{div}$ or $\mathcal{L}_{aln}$ results in a performance decline, validating the effectiveness of the proposed auxiliary losses in enhancing cross-subject generalizability.

## 5.4 Sensitivity Analysis

A series of sensitivity analyses are performed to assess the influence of hyperparameters, including the number of MGRT layer and GRT layer, augmentation ratio, and hidden dimension. The hyperparameters can be determined based on results in Figure 3.

## 5.5 Visualization Analysis

We employ the t-SNE technique [58] to visualize the distribution of both temporal and decomposed features, aiming to gain insights into the feature space. In Figure 4 (a), the temporal features display a dispersed distribution, with a noticeable gap between EEG and PPS. However, within each modality, it is not possible to distinctly separate the concordant and discrepant channels. In contrast, the decomposed features, depicted in Figure 4 (b), exhibit a clearly dispersed distribution with the concordant and discrepant channels distinctly separated, highlighting that the CMMGD can learn concordant and discrepant representations.

In Figure 5, we visualize the activation of each channel in the EEG signals averaged on all samples of the left-out subject during the validation process. The red regions indicate the highly activated brain regions, while the blue regions are less activated. We mark partially consistent highly activated brain regions across subjects with circles, indicating that the proposed CMMGD framework can effectively capture robustness brain activation patterns.

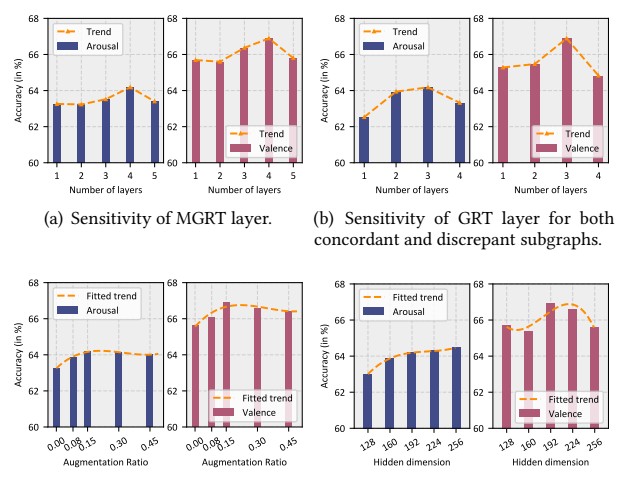

(a) Sensitivity of MGRT layer.

(b) Sensitivity of GRT layer for both concordant and discrepant subgraphs.

(c) Sensitivity of augmentation ratio.

(d) Sensitivity of hidden dimension.

**Figure 3: Sensitivity analysis of hyperparameters.**

**Table 4: Comparison of the proposed MGRT backbone with graph-based networks, transformer, and graph transformers.**

| Type | Backbone | DEAP | | | | | MAHNOB−HCI | | | | |
|---|---|---|---|---|---|---|---|---|---|---|---|
| | | Arousal | | Valence | | Overall Metrics | Arousal | | Valence | | Overall Metrics |
| | | Accuracy | F1 Score | Accuracy | F1 Score | | Accuracy | F1 Score | Accuracy | F1 Score | |
| Single Architecture | GCN [17] | 63.37/10.67 | 70.11/17.11 | 62.48/5.90 | 70.87/7.91 | 66.71 | 64.37/8.77 | 61.31/22.19 | 61.14/6.61 | 70.14/8.51 | 61.74 |
| | GraphConv [40] | 62.19/11.11 | 70.53/16.57 | 63.14/6.84 | 71.41/7.60 | 66.82 | 63.34/10.11 | 62.04/21.64 | 61.42/5.77 | 70.02/8.51 | 64.21 |
| | Transformer [59] | 61.82/11.29 | 71.04/14.67 | 63.04/7.35 | 71.43/7.50 | 66.83 | 62.16/9.90 | 59.76/21.96 | 61.47/6.37 | 70.07/9.17 | 63.37 |
| Graph Transformer | GraphGPS [44] | 63.02/10.92 | **71.21**/15.30 | 61.05/7.54 | 70.95/7.44 | 66.56 | 64.61/11.63 | 62.60/21.56 | 57.03/8.51 | 70.07/9.48 | 63.58 |
| | EmoGTs [14] | 63.62/10.31 | 70.41/16.52 | 64.44/6.64 | 72.16/6.87 | 67.66 | 65.47/10.54 | 62.24/21.30 | 63.02/5.13 | 70.24/9.06 | 65.24 |
| | CMMGD (Ours) | **64.18**/10.15 | 70.75/16.25 | **66.89**/6.34 | **72.55**/6.81 | **68.59** | **66.41**/11.00 | **62.61**/23.19 | **65.86**/7.15 | **72.65**/6.77 | **66.88** |

**Table 5: Ablation study of dropping discrepant features.**

| Dataset | Add $\mathcal{G}_{dis}$ | Arousal | | Valence | | Overall Metrics |
|---|---|---|---|---|---|---|
| | | Accuracy | F1 Score | Accuracy | F1 Score | |
| DEAP | | 61.48/12.37 | **71.63**/14.08 | 63.18/7.19 | 71.23/7.73 | 66.88 |
| | ✓ | **64.18**/10.15 | 70.75/16.25 | **66.89**/6.34 | **72.55**/6.81 | **68.59** |
| MAH-NOB | | 62.96/9.79 | 60.71/23.56 | 60.40/6.52 | 70.98/7.73 | 63.76 |
| | ✓ | **66.41**/11.00 | **62.61**/23.19 | **65.86**/7.15 | **72.65**/6.77 | **66.88** |

**Table 6: Ablation study of varying fusion methods.**

| Dataset | Fusion Method | Arousal | | Valence | | Overall Metrics |
|---|---|---|---|---|---|---|
| | | Accuracy | F1 Score | Accuracy | F1 Score | |
| DEAP | Sum | 59.88/14.71 | 71.27/14.19 | 60.16/7.78 | 70.94/7.76 | 65.56 |
| | Concat | 60.46/13.34 | **71.34**/14.34 | 60.39/8.56 | 71.27/7.98 | 65.87 |
| | Max | 61.96/11.79 | 70.67/15.62 | 61.68/8.82 | 71.94/7.47 | 66.61 |
| | Ours | **64.18**/10.15 | 70.75/16.25 | **66.89**/6.34 | **72.55**/6.81 | **68.59** |
| MAHNOB | Sum | 64.67/10.26 | **63.83**/18.86 | 59.39/9.19 | 69.76/8.63 | 64.41 |
| | Concat | 63.39/11.19 | 60.51/23.91 | 58.48/8.74 | 69.17/10.52 | 62.89 |
| | Max | 64.23/10.91 | 62.28/21.35 | 57.79/8.78 | 69.42/9.38 | 63.39 |
| | Ours | **66.41**/11.00 | 62.61/23.19 | **65.86**/7.15 | **72.65**/6.77 | **66.88** |

## 6 Conclusions

This study proposes the CMMGD framework to effectively confronts the challenges posed by multi-modality heterogeneity and cross-subject divergence in cross-subject multimodal emotion recognition and offers a unified framework that simultaneously mitigates these issues. By decomposing multimodal signals into concordant and discrepant representations, the CMMGD facilitates a comprehensive analysis of the data. A cross-rebalance fusion mechanism is introduced to adaptively fuse each subgraph in a balanced manner. Additionally, CMMGD contains a specifically devised MGRT

**Table 7: Ablation study of the adopted auxiliary loss.**

| Dataset | Auxiliary Loss | | Arousal | | Valence | | Overall Metrics |
|---|---|---|---|---|---|---|---|
| | $\mathcal{L}_{div}$ | $\mathcal{L}_{aln}$ | Accuracy | F1 Score | Accuracy | F1 Score | |
| DEAP | | | 62.47/10.27 | 69.56/16.20 | 63.97/7.79 | 71.06/8.25 | 66.77 |
| | | ✓ | 62.21/10.14 | 70.40/14.31 | 63.70/7.93 | 71.78/7.05 | 67.02 |
| | ✓ | | 63.47/10.19 | 70.01/17.66 | 65.50/6.39 | 72.19/7.69 | 67.79 |
| | ✓ | ✓ | **64.18**/10.15 | **70.75**/16.25 | **66.89**/6.34 | **72.55**/6.81 | **68.59** |
| MAHNOB | | | 61.97/13.38 | 56.32/24.32 | 62.06/5.65 | 70.44/8.70 | 62.70 |
| | | ✓ | 60.66/13.27 | 56.33/23.39 | 62.17/5.91 | 70.00/9.11 | 62.29 |
| | ✓ | | 64.52/10.37 | 60.11/21.98 | 61.67/6.24 | 70.15/9.56 | 64.11 |
| | ✓ | ✓ | **66.41**/11.00 | **62.61**/23.19 | **65.86**/7.15 | **72.65**/6.77 | **66.88** |

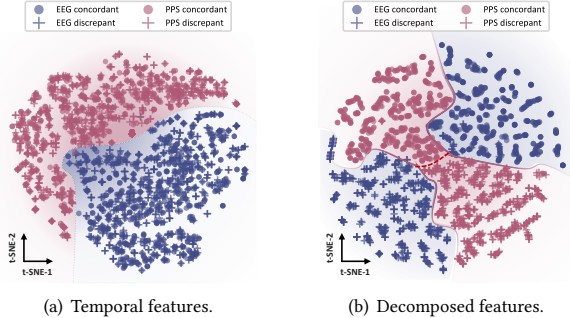

(a) Temporal features.  (b) Decomposed features.

**Figure 4: Channel distribution of features.**

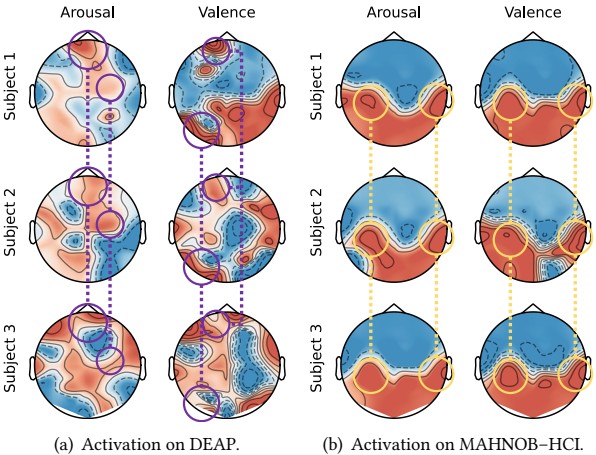

(a) Activation on DEAP.  (b) Activation on MAHNOB−HCI.

**Figure 5: The visualization of brain activation in the DEAP and MAHNOB−HCI datasets.**

backbone that can capture both local and global information in multimodal physiological signals.

Our work presents a promising step towards solving the critical challenge of cross-subject generalization for multimodal contents. One limitation remains the requirement of complete data for each modality. Future work will explore the potential of interpolating missing data, or dealing with noisy data, to enhance the robustness of the CMMGD framework. Moreover, we will investigate the potential of the CMMGD framework in other multimodal tasks.

## Acknowledgments

This work was supported by the National Key Research and Development Plan of China (No. 2023YFC3604802), the Youth Innovation Promotion Association CAS, the Science and Technology Innovation Program of Hunan Province (No. 2022RC4006, No. 2023WK2005), and the Hunan Provincial Natural Science Foundation of China (No. 2023JJ70034, No. 2023JJ70008).

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
