# OpenReview forum: "Correlation-Driven Multi-Modality Graph Decomposition for Cross-Subject Emotion Recognition"
_acmmm.org/ACMMM/2024/Conference — MM2024 Poster_

### Official Review · Reviewer_zV7G · 2024-05-24

**Rating:** 4
**Confidence:** 2

**Summary:**

The paper presents a novel Correlation-Driven Multi-Modality Graph Decomposition (CMMGD) framework for cross-subject emotion recognition using multi-modality physiological signals. CMMGD addresses multi-modality heterogeneity and cross-subject divergence by capturing adaptive cross-modal correlations and decomposing them into concordant and discrepant subgraphs. The methodology is clearly explained with the inclusion of diagrams and pseudocode. Extensive experiments demonstrate the superiority of CMMGD.

**Strengths:**

1. The overall logic of the paper is clear, and the method is well-structured. The inclusion of the main framework and pseudocode adds clarity to the explanation of the proposed method.
2. The experiments are comprehensive and include specific tests that highlight the unique contributions of the proposed method.

**Limitations:**

1. The authors need to compare more extensively with recent works, including discussing them in the section Introduction and considering them as stronger baselines, such as [1] and [2], but we recognize that there are more relevant studies beyond these.
  - [1] Liu, Chenyu, Xinliang Zhou, Zhengri Zhu, Liming Zhai, Ziyu Jia, and Yang Liu. "VBH-GNN: Variational Bayesian Heterogeneous Graph Neural Networks for Cross-subject Emotion Recognition." In The Twelfth International Conference on Learning Representations.

  - [2] Zhong, Xiaolong, Fei Wu, Zhong Yin, and Gang Liu. "An Attention-Enhanced Retentive Broad Learning System for Subject-Generic Emotion Recognition from EEG Signals." In ICASSP 2024-2024 IEEE International Conference on Acoustics, Speech and Signal Processing (ICASSP), pp. 2310-2314. IEEE, 2024.

2. Have the authors considered the impact of individual differences on consistency and inconsistency?

3. During the decomposition of the multi-modality mixed graph, how do the authors ensure that the resulting subgraphs accurately represent concordant and discrepant features?

**Suitability:**

3

---

### Official Review · Reviewer_UWAX · 2024-05-24

**Rating:** 5
**Confidence:** 4

**Summary:**

The paper introduces CMMGD as a new unified framework that initially captures adaptive cross-modal correlations between different modalities like EEG and PPS. It then decomposes these multimodal signals into concordant and discrepant subgraphs based on the learned correlations. This decomposition is intended to isolate consistently activated features across modalities and subjects during emotion elicitation, which are believed to form a generalizable subspace. Additionally, the study develops a Multi-Modality Graph Regularized Transformer (MGRT) backbone specifically tailored for multimodal physiological signals, which helps mitigate over-smoothing and avoid over-reliance on any single modality, thereby promoting generalizability.

**Strengths:**

1. The proposed CMMGD framework offers a fresh perspective on addressing multimodal heterogeneity and cross-subject divergence by decomposing multimodal graphs. This approach allows for a more nuanced analysis of emotional states compared to traditional fusion methods.
2. The use of a graph transformer architecture with local and global attention mechanisms within the MGRT backbone seems well-suited to capture both local and global features from multimodal data, potentially improving model performance.
3. The introduction of an alignment loss and a divergence loss further enhances the model's ability to generalize across subjects, which is a significant advantage for real-world applications where subject variability is high.

**Limitations:**

The conceptual framework is complex, making it difficult for readers outside the field of machine learning and signal processing to grasp all the nuances involved in the methodology.

**Suitability:**

3

---

### Official Review · Reviewer_1Sua · 2024-05-24

**Rating:** 4
**Confidence:** 4

**Summary:**

The paper presents a novel framework, CMMGD, addressing multi-modality heterogeneity and cross-subject divergence in the context of emotion recognition from physiological signals. The authors propose a correlation-driven approach to decompose multimodal signals into concordant and discrepant subgraphs, aiming to capture consistent and variable patterns across subjects. The framework includes a specialized backbone, MGRT, designed to handle the unique challenges of small-scale physiological graph data. The CMMGD framework demonstrates improved performance over existing methods on the DEAP and MAHNOB-HCI datasets.

**Strengths:**

(1) The CMMGD framework innovatively addresses the dual challenge of multi-modality heterogeneity and cross-subject divergence, which are critical issues in cross-subject emotion recognition.
(2) The introduction of the MGRT backbone is a significant contribution, providing a specialized solution for handling multimodal physiological signals and mitigating over-smoothing.
(3) The paper includes a comprehensive set of experiments and ablation studies that thoroughly validate the effectiveness of the proposed framework and its components.
(4) The cross-rebalance fusion mechanism is a thoughtful addition that adaptively fuses concordant and discrepant subgraphs, potentially leading to more accurate emotion recognition.
(5) The authors provide a clear and detailed explanation of the methodology, along with a well-structured presentation of experimental results and analyses.

**Limitations:**

(1) The paper lacks a detailed discussion on the computational complexity and efficiency of the proposed framework, which is crucial given the graph-based nature of the approach.
(2) The theoretical justification for certain design choices, such as the specific architecture of the MGRT backbone or the choice of auxiliary losses, could be more rigorously explained.
(3) The paper could benefit from a more thorough comparison with state-of-the-art methods, particularly those that also aim to address cross-subject generalization in emotion recognition.
(4) The authors should provide more insight into how the framework handles missing or incomplete data, which is often a concern in real-world physiological signal datasets.
(5) The paper does not sufficiently address potential limitations or the robustness of the framework to noisy data, which is an important consideration for practical applications.

**Suitability:**

3

---

### Meta-Review · Area_Chair_iyXv · 2024-06-26

**Recommendation:** Accept (Poster)
**Confidence:** 5

**Metareview:**

Based on the ratings of reviewers, I suggest to accept the paper .

---

### Meta-Review · Senior_Area_Chairs · 2024-07-10

**Recommendation:** Accept (Poster)
**Confidence:** 4

**Metareview:**

All the reviewers gave positive ratings and tend to accept the paper. SAC and AC agree with reviewers and recommend acceptance of the paper.